# Fundamental Relation for the Ideal Gas in the Gravitational Field and Heat Flow

**DOI:** 10.3390/e25111483

**Published:** 2023-10-26

**Authors:** Robert Hołyst, Paweł J. Żuk, Karol Makuch, Anna Maciołek, Konrad Giżyński

**Affiliations:** 1Institute of Physical Chemistry, Polish Academy of Sciences, Kasprzaka 44/52, 01-224 Warszawa, Poland; kmakuch@ichf.edu.pl (K.M.); amaciolek@ichf.edu.pl (A.M.); kgizynski@ichf.edu.pl (K.G.); 2Department of Physics, Lancaster University, Lancaster LA1 4YB, UK; 3Max-Planck-Institut für Intelligente Systeme Stuttgart, Heisenbergstr. 3, D-70569 Stuttgart, Germany

**Keywords:** thermodynamics, non-equilibrium thermodynamics, gravity, stationary-state, steady-state, entropy

## Abstract

We formulate the first law of global thermodynamics for stationary states of the ideal gas in the gravitational field subjected to heat flow. We map the non-uniform system (described by profiles of the density and temperature) onto the uniform one and show that the total internal energy $U(S^{*},V,N,L,M^{*})$ is the function of the following parameters of state: the non-equilibrium entropy $S^{*}$, volume *V*, number of particles, *N*, height of the column *L* along the gravitational force, and renormalized mass of a particle $M^{*}$. Each parameter corresponds to a different way of energy exchange with the environment. The parameter $M^{*}$ changes internal energy due to the shift of the centre of mass induced by the heat flux. We give analytical expressions for the non-equilibrium entropy $S^{*}$ and effective mass $M^{*}$. When the heat flow goes to zero, $S^{*}$ approaches equilibrium entropy. Additionally, when the gravitational field vanishes, our fundamental relation reduces to the fundamental relation at equilibrium.

## 1. Introduction

Classical equilibrium thermodynamics describes the energy exchange between the system of interest and the rest of the world called the environment. An important result of this theory is the fundamental relation, which gives the internal energy, *U*, as a function of parameters of state [1,2]. Each state parameter represents one independent way of energy exchange with the environment. This fundamental relation is sufficient to fully describe all thermodynamic properties of the system and all thermodynamic reversible processes taking the system from one equilibrium state to the other. Until recently, such a description for systems in non-equilibrium states has not been available.

Non-equilibrium states are characterized by macroscopic energy fluxes flowing across the system. These fluxes are possible only if the system is non-uniform with non-vanishing gradients of temperature (in the case of heat flow) or pressure (in the case of mass flow). In our recent contributions, we have found fundamental relations for three different systems in non-equilibrium stationary states: the ideal gas in a heat flow [3], van der Waals gas in the heat flow [4] and a binary mixture of ideal gases in the heat flow [5]. Here, we show the fundamental relation for the ideal gas subjected to the heat flow and gravitational field. Thus, the previous analysis is extended to the non-equilibrium system in the external potential.

The characteristic feature of the above global thermodynamics in stationary states is that the fundamental relation is formally the same as in equilibrium but with additional parameters of state. We approach the construction of non-equilibrium thermodynamics as follows. First, we identify equilibrium equations of state locally in the non-uniform system. Next, we average these equations of state over the system’s volume. After averaging, we obtain a global equation of state, which we write in the same form as at equilibrium. However, it includes new state parameters. Under such construction, material parameters in the equilibrium equations of state become parameters of state in the non-equilibrium state of the system. For example, in the case of van der Waals gas in a heat flow, $U(S^{*},V,N,a^{*},b^{*})$ is a function of 5 parameters of state: the entropy $S^{*}$, volume *V*, number of particles *N*, and the rescaled van der Waals parameters $a^{*}$, $b^{*}$. The state parameters, $a^{*}$, $b^{*}$, together with $S^{*}$, determine the net heat exchange with the environment. The same formal form of *U* is valid as in the equilibrium system [2,4]:(1)$$U=N{\left(\frac{V}{N}-b^{*}\right)}^{-\frac{1}{c}}\exp\left[\frac{S^{*}-Ns_{0}}{cNk_{B}}\right]-a^{*}\frac{N^{2}}{V},$$
In the above, the number of degrees of freedom of motion per particle is *c*, $s_{0}$ is a constant and $k_{B}$ is the Boltzmann constant. For a system with a constant number of particles, we get the differential of *U*
(2)$$dU=T^{*}dS^{*}-pdV-\frac{N^{2}}{V}da^{*}+Nk_{B}T^{*}{\left(\frac{V}{N}-b^{*}\right)}^{-1}db^{*}.$$
From this equation, the net heat flowing in/out of the system and changing the internal energy can be identified as:(3)$$\;d\;\bar{}\;\;Q=T^{*}dS^{*}-\frac{N^{2}}{V}da^{*}+Nk_{B}T^{*}{\left(\frac{V}{N}-b^{*}\right)}^{-1}db^{*}.$$
The $a^{*}$ and $b^{*}$ appear in the net heat differential because, in a non-uniform system, the change in the density profile generally leads to heat absorption or release. The exception is a process that deliberately keeps $\;d\;\bar{}\;\;Q=0$ while parameters $S^{*},a^{*},b^{*}$ change appropriately.

For the ideal gas in equilibrium in the gravitational field, internal energy U(S,V,N,L) is a function of the entropy *S*, the volume *V*, the number of particles *N* and the linear size of the system *L* along the direction of the gravitational force. In this contribution, we derive an analogous relation for the same gas column but subjected to the heat flow. The other purpose of this work is to formulate the first law of global thermodynamics for an ideal gas in the external field and the heat flow.

## 2. The Irreversible Hydrodynamics Approach

We consider the gas column with total height *L*, spanned in *z* direction from the base at $z_{0}$ to the top at $z_{L}$ ($L=z_{L}-z_{0}$) (Figure 1). In this study, we elucidate the quiescent stationary situation, where quiescent means a state with no macroscopic motion of the gas. Therefore, we restrict to the situation with the gravity field g=(0,0,−g) oriented along the direction of the heat flux going between the base of the column (kept at constant temperature $T_{0}$) and the top of the column (kept at constant temperature $T_{L}$). In the case of misalignment, there is no quiescent stationary state in a compressible fluid [6,7]. In alignment, the quiescent state is possible as long as the Rayleigh-Bennard convection [8,9,10,11,12] is absent. As we restrict to the quiescent state only, the energy transport results from the heat conduction in quiescent gas following Fourier’s law [13], which leads to the diffusion equation for the temperature profile. In such conditions, we observe the translational symmetry in x,y directions. However, in contrast to the case of perfect gas [3] and interacting gas [4] in heat flow, the external field removes translational invariance from the pressure profile. Thus, we consider the column that has specified surface area *A* of the column base and top to point out that performing work at various column positions, be it top, bottom or sides, has distinguishable energetic consequences.

The dimensioned conservation laws of irreversible hydrodynamics for the momentum and energy for a monatomic ideal gas in stationary (no temporal changes) and quiescent v = **0** (zero velocity) conditions are [14]
(4a)$$\begin{array}{c} \partial_{z}p=-g\rho , \end{array}$$
(4b)$$\begin{array}{c} \partial_{z}^{2}T=0, \end{array}$$
(4c)$$\begin{array}{c} p=\frac{k_{B}}{M}\rho T, \end{array}$$
(4d)$$\begin{array}{c} u=\frac{3k_{B}}{2M}T \end{array}$$
where ρ is gas density, *p* is thermodynamic pressure, *T* is temperature, $k_{B}$ is Boltzmann’s constant, *M* is the mass of the gas molecule, and *u* is internal energy density per unit mass. The Equation (4b) is a second-order linear ordinary differential equation. For the fixed value boundary conditions ($T_{0}$ at the base and $T_{L}$ at the top), the following linear solution is found [15]
(5)$$T\left(z\right)=T_{0}+\left(T_{L}-T_{0}\right)\frac{z-z_{0}}{L}.$$
Substitution of density obtained from (4c) into (4a) leads to the differential equation for the pressure profile
(6)$$\partial_{z}p=-g\frac{M}{k_{B}}\frac{p}{T}.$$
It has the following solution
(7)$$p=p_{0}\exp\left[-\frac{MgL}{k_{B}T_{0}}\int_{z_{0}/L}^{z/L}\frac{dz^{\prime }}{1+\left(\frac{T_{L}}{T_{0}}-1\right)\frac{z^{\prime }-z_{0}}{L}}\right]=p_{0}{\left(1+\left(\frac{T_{L}}{T_{0}}-1\right)\frac{z-z_{0}}{L}\right)}^{-\frac{MgL}{k_{B}\left(T_{L}-T_{0}\right)}},$$
where $p_{0}=p\left(z=z_{0}\right)$ is the pressure at the ground level. The value of $p_{0}$ can be calculated utilizing the constraint on the total mass present in the system
(8)$$NM=\int_{z_{0}/L}^{z_{L}/L}AL\frac{Mp}{k_{B}T}dz^{\prime }=p_{0}\frac{A}{g}\left(1-{\left(1+(\frac{T_{L}}{T_{0}}-1)\right)}^{-\frac{MgL}{k_{B}\left(T_{L}-T_{0}\right)}}\right),$$
which gives a pressure profile
(9)$$p=\frac{NMg}{A}\frac{{\left(1+\left(\frac{T_{L}}{T_{0}}-1\right)\frac{z-z_{0}}{L}\right)}^{-\frac{MgL}{k_{B}\left(T_{L}-T_{0}\right)}}}{1-{\left(1+(\frac{T_{L}}{T_{0}}-1)\right)}^{-\frac{MgL}{k_{B}\left(T_{L}-T_{0}\right)}}}.$$
In the above, a barometric formula can be recognized. Specifically in the form extended to linear temperature profile [16,17]. Such a pressure distribution has a corresponding density profile
(10)$$\rho =\frac{Mp}{k_{B}T}=\frac{NMg}{A}\frac{M}{k_{B}T_{0}}\frac{{\left(1+\left(\frac{T_{L}}{T_{0}}-1\right)\frac{z-z_{0}}{L}\right)}^{-\frac{MgL}{k_{B}\left(T_{L}-T_{0}\right)}-1}}{1-{\left(1+(\frac{T_{L}}{T_{0}}-1)\right)}^{-\frac{MgL}{k_{B}\left(T_{L}-T_{0}\right)}}}.$$

## 3. Energies

We apply the mapping of a non-uniform system into a uniform one (see [3,4]) to find new state parameters. The solution of local conservation laws contains complete information about the gas column system. Therefore, we are able to calculate energy present in the system for a given set of parameters $\left(T_{0},T_{L},z_{0},z_{L},A,M,N,g\right)$. There are two distinct forms of energy in the system: the energy of the thermal motion of particles and the gravitational energy of particles, which is also referred to as potential energy. The total thermal energy is given by
(11)$$E_{T}=AL\int_{z_{0}/L}^{z_{L}/L}\frac{3k_{B}}{2M}\rho Tdz^{\prime }=\frac{3}{2}AL\int_{z_{0}/L}^{z_{L}/L}pdz^{\prime }=\frac{3}{2}NMgL\frac{1+\frac{\left(\frac{T_{L}}{T_{0}}-1\right)}{1-{\left(1+(\frac{T_{L}}{T_{0}}-1)\right)}^{\frac{MgL}{k_{B}\left(T_{L}-T_{0}\right)}}}}{\frac{MgL}{k_{B}T_{0}}-\frac{T_{L}}{T_{0}}+1},$$
which we use to define a renormalized temperature of the column
(12)$$E_{T}=\frac{3}{2}Nk_{B}T^{*},\;T^{*}=\frac{MgL}{k_{B}}\frac{1+\frac{\left(\frac{T_{L}}{T_{0}}-1\right)}{1-{\left(1+(\frac{T_{L}}{T_{0}}-1)\right)}^{\frac{MgL}{k_{B}\left(T_{L}-T_{0}\right)}}}}{\frac{MgL}{k_{B}T_{0}}-\frac{T_{L}}{T_{0}}+1}.$$
The gravitational potential energy above the ground level, which for clarity, we set to
(13)$$E_{\mathrm{pot},z_{0}}=0$$
is
(14)$$E_{\mathrm{pot}}-E_{\mathrm{pot},z_{0}}=E_{\mathrm{pot}}=gAL^{2}\int_{z_{0}/L}^{z_{L}/L}\rho z^{\prime }dz^{\prime }=NMgL\frac{1+\left(\frac{\frac{MgL}{k_{B}T_{0}}}{1-{\left((\frac{T_{L}}{T_{0}}-1)+1\right)}^{\frac{MgL}{k_{B}\left(T_{L}-T_{0}\right)}}}\right)}{\frac{MgL}{k_{B}T_{0}}-\frac{T_{L}}{T_{0}}+1}.$$
We use the total gravitational energy to define the renormalized mass of the system $M^{*}$
(15)$$E_{\mathrm{pot}}=\frac{NM^{*}gL}{2},\;M^{*}=2M\frac{1+\left(\frac{\frac{MgL}{k_{B}T_{0}}}{1-{\left((\frac{T_{L}}{T_{0}}-1)+1\right)}^{\frac{MgL}{k_{B}\left(T_{L}-T_{0}\right)}}}\right)}{\frac{MgL}{k_{B}T_{0}}-\frac{T_{L}}{T_{0}}+1}.$$
In such a way, we obtained the new state parameter resulting from the mapping procedure. For the discussion of the meaning of $E_{\mathrm{pot},z_{0}}\neq 0$ see the Appendix A.

In Figure 2 we illustrate the general behavior of $E_{T}$ and $E_{\mathrm{pot}}$.

The gas in the column is subject to gravitational field and heat flux that is, subject to two distinct forcings. As a result, more than one limiting transition exists.

The first one is when heat flux ceases $T_{L}=T_{0}$ and we obtain $T^{*}=T_{0}$. Regardless of the strength of the gravitational field, we find that $E_{T}=\frac{3}{2}Nk_{B}T$ (Figure 2a). However, $M^{*}$ is not equal to *M* (Figure 2b) and the potential energy depends on the strength of the gravitational field. For high $T_{L}/T_{0}$, thermal energy becomes linear as a function of $T_{L}$, but the exact value depends on the gravitational field (Figure 2a). Stronger fields tend to keep more gas close to the colder (bottom) wall. The effect similar to this for stronger gravitational fields is visible at low $T_{L}$. More gas resides close to the warmer (bottom) wall. The potential energy is strongly influenced by the temperature profile for low $T_{L}/T_{0}$ (Figure 2b), but as the temperature grows, it reaches a constant value proportional to the strength of the gravitational field.

The second limiting transition happens when g=0. We recover exactly the $T^{*}$ corresponding to the ideal gas in heat flux as presented by Hołyst et al. [3] (Figure 2c). For small gravitational fields in comparison to the thermal energy, but not equal to 0, the gravitational energy contains the linear term only and $M^{*}=M$ (Figure 2d). This represents the situation of incompressible gas or fluid. When gravity becomes dominant, the thermal energy converges to the one appropriate for the bottom wall (Figure 2c). Most of the gas will reside in its vicinity. Interestingly, for the potential energy, there is a limiting value equal $Nk_{B}T_{0}$ (2/3 in Figure 2d) for all $T_{L}$. It can be rationalized in the following way: the increase in the strength of the gravitational field lowers the centre of mass. Therefore, the gravitational energy inside the column cannot grow indefinitely.

## 4. Equations of State

In classical thermodynamics, equations of state like
(16)$$p=\frac{Nk_{B}T}{V},\;U=\frac{3Nk_{B}T}{2}$$
that describe the ideal gas are necessary to write the specific form of fundamental relation. Likewise, global stationary thermodynamics needs its analogues. In a system with potential gravitational energy imposed by an external field, the total energy stored in the system consists of both thermal and potential energies
(17a)$$U=E_{T}+E_{\mathrm{pot}}=\frac{3}{2}Nk_{B}T^{*}+\frac{NM^{*}gL}{2}.$$
Additionally, the following relations between functional forms for pressure and energy components hold
(17b)$$p_{\mathrm{av}}=\frac{2}{3AL}E_{T}=\frac{Nk_{B}T^{*}}{AL}$$
(17c)$$p\left(z_{L}\right)=\frac{2}{3AL}\left(E_{T}-\frac{3}{2}E_{\mathrm{pot}}\right)=\frac{1}{AL}\left(Nk_{B}T^{*}-\frac{NM^{*}gL}{2}\right),$$
which can be checked by substitution. Also, AL=V can be substituted to obtain a functional form even closer to (Equation 16). Together, Equations (17) form a set of effective equations of state for a column of a perfect gas in the gravitational field.

Similar effective equations of state exist for the interacting Van der Waals gas [4]. The difference is that here, pressure is not uniform, and that total internal energy *U* is increased by the external gravitational potential while the potential energy of particle-particle interactions diminish it.

## 5. Fundamental Relation

For equilibrium, in the special case of constant temperature $T^{*}=T_{0}=T_{L}$, constant number of particles *N*, and $\frac{MgL}{k_{B}T_{0}}\ll 1$ (yields $M^{*}=M$) one obtains the following fundamental relation
(18)$$U=\left(U_{0}-\frac{NMgL_{0}}{2}\right){\left(\frac{V}{V_{0}}\right)}^{-2/3}\exp\left[\frac{2}{3}\frac{S-S_{0}}{Nk_{B}}\right]+\frac{NMgL}{2},$$
and its the equivalent form
(19)$$S=\frac{3}{2}Nk_{B}\ln\frac{U-\frac{NMgL}{2}}{U_{0}-\frac{MgL_{0}}{2}}+Nk_{B}\ln\frac{V}{V_{0}}+S_{0}.$$
The reference values noted by $U_{0}$ for internal energy, $S_{0}$ for entropy, $L_{0}$ for the column’s height and $V_{0}$ for the column’s volume. A conceptually similar fundamental relation, but for a system symmetric with respect to the $z_{0}=0$ i.e., z∈(−L,L) can be found in [18]. There are also approaches to formulating the fundamental relation stemming from the microscopic arguments and using statistical mechanics approaches [19,20], but these are limited to constant temperature conditions.

Equations of state ([Disp-formula FD17a-entropy-25-01483])–([Disp-formula FD17c-entropy-25-01483]) for the system with heat flow are formally similar to the equations of state of an ideal gas in the gravity field in equilibrium. Motivated by our previous works, where we showed that non-equilibrium systems can be mapped to their equilibrium counterpart by introducing effective parameters such as non-equilibrium entropy $S^{*}$ and effective interaction parameter $a^{*}$ in Equation (Equation 1) we explore in this article whether the fundamental relation of the form (Equation 18) is satisfied. In what follows, we postulate the following fundamental relation
(20)$$U=\left(U_{0}-\frac{NM_{0}gL_{0}}{2}\right){\left(\frac{V}{V_{0}}\right)}^{-2/3}\exp\left[\frac{2}{3}\frac{S^{*}-S_{0}}{Nk_{B}}\right]+\frac{NM^{*}gL}{2},$$
with parameters of state $S^{*},V,L,N,M^{*}$ and $U_{0}$, $S_{0}$, $V_{0}$, $L_{0}$, and $M_{0}$ being reference constants.

By definition, the fundamental relation contains the whole thermodynamic information about the system. In particular, it is possible to get the equations of state ([Disp-formula FD17a-entropy-25-01483])–([Disp-formula FD17c-entropy-25-01483]). It is straightforward to check that it is indeed the case. For example
(21)$${\left.\frac{\partial U}{\partial S^{*}}\right|}_{A,N,L,M^{*}}=\frac{2}{3Nk_{B}}\left(U-\frac{NM^{*}gL}{2}\right)=\frac{2E_{T}}{3Nk_{B}}=T^{*},$$
$${\left.-\frac{1}{A}\frac{\partial U}{\partial L}\right|}_{S^{*},A,N,M^{*}}=\frac{1}{AL}\left(Nk_{B}T^{*}-N\frac{M^{*}gL}{2}\right)=p\left(z_{L}\right),$$
and
$${\left.-\frac{1}{L}\frac{\partial U}{\partial A}\right|}_{S^{*},A,N,M^{*}}=\frac{Nk_{B}T^{*}}{AL}=p_{av}.$$
Similar to the equilibrium case, there is an equivalent representation of fundamental relation (Equation 20) in the form of entropy
(22)$$S^{*}=\frac{3}{2}Nk_{B}\ln\frac{U-\frac{NM^{*}gL}{2}}{U_{0}-\frac{NM_{0}gL_{0}}{2}}+Nk_{B}\ln\frac{V}{V_{0}}+S_{0}$$

## 6. Work

Another property of the fundamental relation is that it allows one to identify heat and work performed on the system. They appear as a result of a transition from one stationary state to another due to the change of the control parameters $T_{0},T_{L},z_{0},z_{L},A$. The infinitesimal change in total internal energy stored in the column is due to the excess heat flowing into the column and due to the work exerted on the column’s boundaries [3]
(23)$$dU=\;d\;\bar{}\;\;Q+\;d\;\bar{}\;\;W.$$
The work can be exerted on the column in three ways that require different forces to overcome. First, by displacing the base of the column (dz at $z=z_{0}$). The pressure acting at the base of the column is
(24)$$p\left(z_{0}\right)=p\left(z_{L}\right)+\frac{NMg}{A}=\frac{NMg}{A}\frac{1}{1-{\left(1+(\frac{T_{L}}{T_{0}}-1)\right)}^{-\frac{MgL}{k_{B}\left(T_{L}-T_{0}\right)}}}.$$
Although in the main text, we do not discuss in detail the consequences of the floor motion, we elucidate them in the Appendix A. Second, by displacing the upper wall of the system (dz at $z=z_{L}$)
(25)$$p\left(z_{L}\right)=p\left(z_{0}\right)-\frac{NMg}{A}=\frac{NMg}{A}\frac{{\left(1+(\frac{T_{L}}{T_{0}}-1)\right)}^{-\frac{MgL}{k_{B}\left(T_{L}-T_{0}\right)}}}{1-{\left(1+(\frac{T_{L}}{T_{0}}-1)\right)}^{-\frac{MgL}{k_{B}\left(T_{L}-T_{0}\right)}}}.$$
Interestingly, if one displaces both walls in the same way $dz_{0}=dz_{L}$, like in the elevator, they have to overcome the force of gravity acting on the whole column, which equals NMg. In such action, there are no changes inside the gas column, although it is entirely displaced. We provide a more detailed discussion in Appendix A. The third method is to change the surface area dA of the column base and top but keep the height of the column constant. The force to overcome results from averaging pressure over the whole height of the column
(26)$$p_{\mathrm{av}}=\int_{z_{0}/L}^{z_{L}/L}pdz^{\prime }=\frac{NMg}{A}\frac{1+\frac{\left(\frac{T_{L}}{T_{0}}-1\right)}{1-{\left(1+(\frac{T_{L}}{T_{0}}-1)\right)}^{\frac{MgL}{k_{B}\left(T_{L}-T_{0}\right)}}}}{\frac{MgL}{k_{B}T_{0}}-\frac{T_{L}}{T_{0}}+1}.$$
In summary, the infinitesimal work exerted on the system can be written as
(27)$$\;d\;\bar{}\;\;W=Ap\left(z_{0}\right)dz_{0}-Ap\left(z_{L}\right)dz_{L}-Lp_{\mathrm{av}}dA$$
where the first term has an opposite sign due to the orientation of the *z* axis with respect to the body of gas inside the column. For the sake of clarity of the following presentation, we will assume that $z_{0}=0$ and is fixed, which leads to a simplified expression for the infinitesimal work ($dz_{L}=dL$)
(28)$$\;d\;\bar{}\;\;W=-Ap\left(z_{L}\right)dL-Lp_{\mathrm{av}}dA.$$
With Equations ([Disp-formula FD17b-entropy-25-01483]) and ([Disp-formula FD17c-entropy-25-01483]) we get the following equivalent expression
(29)$$\;d\;\bar{}\;\;W=-\frac{N}{L}\left(k_{B}T^{*}-\frac{M^{*}gL}{2}\right)dL-\frac{Nk_{B}T^{*}}{A}dA.$$

## 7. Heat

Finally, we elucidate the functional form of excess heat differential $\;d\;\bar{}\;\;Q$. In the energy balance (Equation 23), we substitute (Equation 29) for $\;d\;\bar{}\;\;W$ and use partial derivatives of the fundamental relation (Equation 20)
(30)$$\begin{array}{cc} dU= & {\left.\frac{\partial U}{\partial S^{*}}\right|}_{M^{*},A,L,N}dS^{*}+{\left.\frac{\partial U}{\partial M^{*}}\right|}_{S^{*},A,L,N}dM^{*}+{\left.\frac{\partial U}{\partial L}\right|}_{S^{*},A,N,M^{*}}dL+{\left.\frac{\partial U}{\partial A}\right|}_{S^{*},L,N,M^{*}}dA\\ = & T^{*}dS^{*}+N\frac{gL}{2}dM^{*}-\frac{N}{L}\left(k_{B}T^{*}-\frac{M^{*}gL}{2}\right)dL-\frac{Nk_{B}T^{*}}{A}dA\\ = & T^{*}dS^{*}+N\frac{gL}{2}dM^{*}+\;d\;\bar{}\;\;W. \end{array}$$
As a result, we obtain
(31)$$\;d\;\bar{}\;\;Q=T^{*}dS^{*}+N\frac{gL}{2}dM^{*}.$$

## 8. Discussion

Global non-equilibrium thermodynamics describes the system’s energy exchange with its environment using a few global state parameters. We have presented such a description for the ideal gas in the gravitational field, subjected to the heat flux in the stationary state. The state parameters for such a system are the non-equilibrium entropy, $S^{*}$, the volume, *V*, the number of particles, *N*, the system size along the gravitational force, *L*, and the effective mass, $M^{*}$. The internal energy is the function of these parameters $U(S^{*},V,N,L,M^{*})$, irrespective of the size of the system, the number of boundary parameters, the shape of the system, and the heat flux. We have given analytical forms of $S^{*}$ and $M^{*}$ for arbitrary strong gravity and heat flux in the case of a gas column. As a limiting case, we have given the exact formulas for the system in the gravitational field but without a heat flux. All our formulas reduce to the equilibrium form of the internal energy when the heat flux goes to zero. In previous contributions, we have formulated the same global thermodynamic description for the ideal gas [3], van der Waals gas [4], and ideal gas mixture [5] in the heat flux. The current development regards the presence of an external potential field. In all studied cases, we observed that the material parameters such as interaction parameters (van der Waals gas), the difference between the number of degrees of freedom (gas binary mixtures), or mass (in the gravitational field) became parameters of state in the heat flow. We expect the same behaviour in other non-equilibrium systems, e.g., in the system of dipoles, the electric susceptibility would become the state parameter. In the van der Waals gas, the additional state parameter and the non-equilibrium entropy describe the net heat absorbed or released by the system in any process. In the van der Waals gas, the additional state parameter and the non-equilibrium entropy describe the net heat absorbed or released by the system in any process. In our case, the net heat is described by the non-equilibrium entropy and the effective mass parameter. The effective mass additionally describes the change in the distribution of density and shift of the location of the centre of mass at constant gravity. The existence of the global thermodynamic description with a well-defined first law of thermodynamics raises hopes that the second law, which predicts the direction of spontaneous processes, might as well, be expressed within the same conceptual framework.

## Figures and Tables

**Figure 1 entropy-25-01483-f001:**
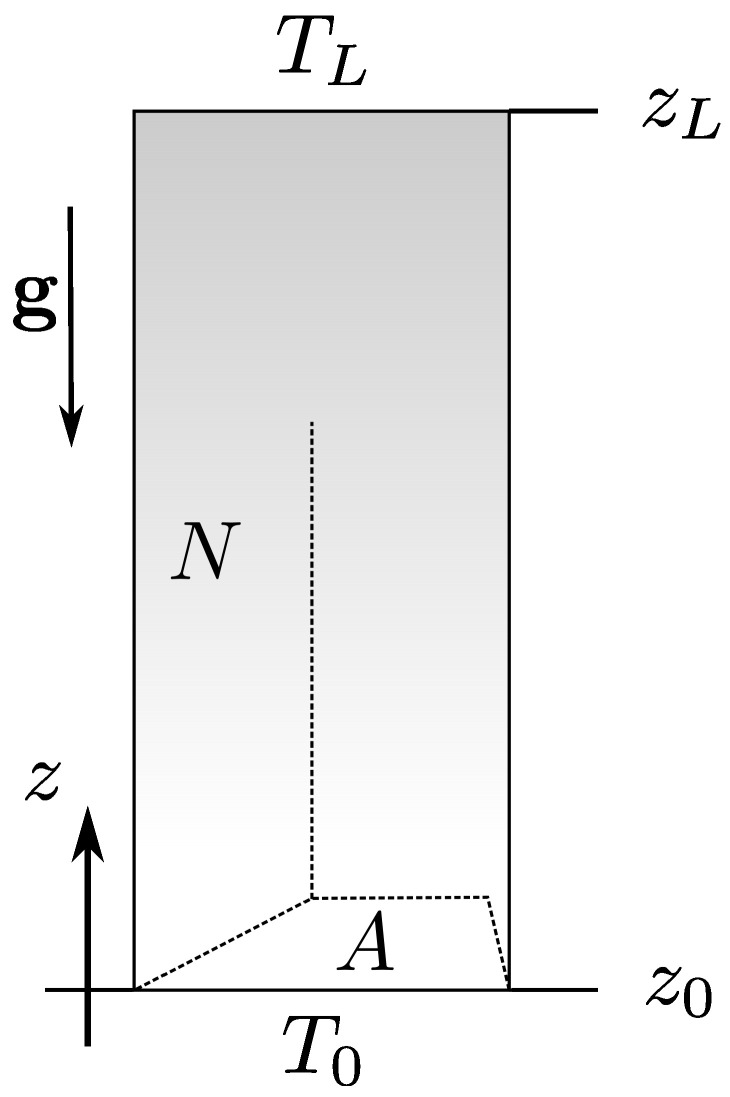
The geometry. A column of height *L* containing *N* gas particles in a gravitational field aligned in the *z* direction. The base ($z_{0}$) and top ($z_{L}$) are in contact with temperature reservoirs kept at two different temperatures: $T_{0}$ at the base and $T_{L}$ at the top. The column base and top surface area equals *A*.

**Figure 2 entropy-25-01483-f002:**
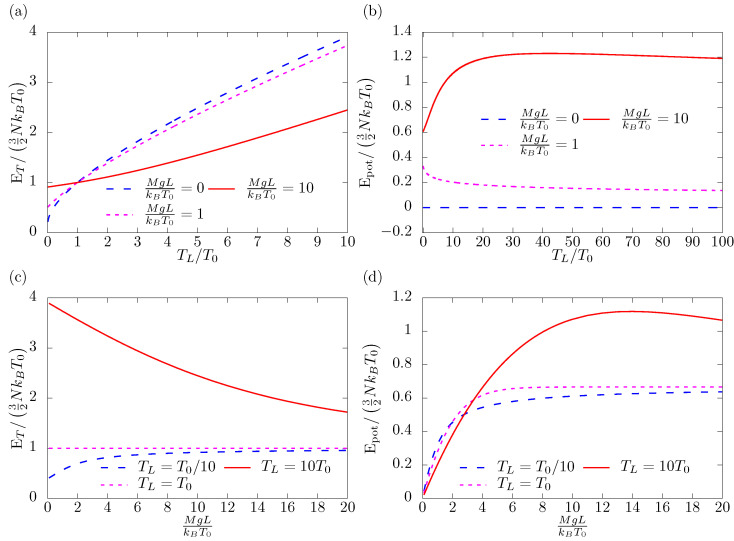
Energies. As a measure of the strength of the gravitational field the ratio of the gravitational and thermal energies of the single particle $\frac{MgL}{RT_{0}}$ is used. (**a**) Thermal energy as a function of $T_{L}$ for three strengths of the gravitational field. (**b**) Potential energy as a function of $T_{L}$ for three strengths of the gravitational field. (**c**) Thermal energy as a function of the strength of gravitational field for three values of $T_{L}$. (**d**) Potential energy as a function of the strength of gravitational field for three values of $T_{L}$.

## Data Availability

No new data were created or analyzed in this study. Data sharing is not applicable to this article.

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
