# Peer review of "Fundamental Relation for the Ideal Gas in the Gravitational Field and Heat Flow"

_entropy, 2023, doi:10.3390/e25111483_

Round 1

Reviewer 1 Report

The work Fundamental relation for the ideal gas in the gravitational field and heat flow” by Robert Holyst et al. continues the series of the works by the authors on the fundamental relations for non-equilibrium settings. The approach is based on a finding that the fundamental relation representing the First Law for a non-equilibrium system following after volume averaging may be casted into the same form as the corresponding equilibrium relation. The price to pay is that some fixed properties of an equilibrium system get to be variables of state for the non-equilibrium one. This is an interesting and well-written work, which can be accepted in its present form. However, there is a subtle point which Authors should comment on in somewhat more detail.

While in the previously considered examples changing parameters of the system for variables of state is more or less harmless, in the new example of a quiescent gas it is less so. The renormalization of the (gravitational) mass leads essentially to the existence of two different gravitational masses, cf. the discussion on line 158, the total and the effective one. Is renormalization of the mass the only way to implement the scheme? A more physical way seems to be an introduction of a new state variable describing the position of the center of mass of the column (is this possible?). This follows the same idea of getting the equation of state with additional parameters of state, but the corresponding parameter is not a previous parameter of the initial system, but a new one. How important is it to has exactly the mass as a state variable? What are the advantages (the disadvantage is evident)?

Reviewer 2 Report

Review of an article entitled Fundamental relation for the ideal gas in the gravitational field and heat flow

1.      The first comment concerns the English language. I believe, although this is already an individual matter, but please use the passive voice more often than the active voice.

2.      There are no references to the literature regarding, for example, formulas 1, 2 and 3.

3.      I understand that in gas only diffusion is considered?

4.      Why was a linear temperature profile assumed in Equation 5? This requires an explanation?

5.      It is hard for me to check some of the transformations and derivations of formulas. Is it possible to show them, for example, in the appendices?

6.      Second major comment is that it is possible to show trends and relationships in graphs. Showing how it relates to equilibrium thermodynamics, etc.

I think there should be some language corrections. The generalities in the use of the passive voice. 
